# LinkNet: Relational Embedding for Scene Graph

**Sanghyun Woo**\*\*
EE, KAIST
Daejeon, Korea
shwoo93@kaist.ac.kr

**Dahun Kim\***
EE, KAIST
Daejeon, Korea
mcahny@kaist.ac.kr

**Donghyeon Cho**
EE, KAIST
Daejeon, Korea
cdh12242@gmail.com

**In So Kweon**
EE, KAIST
Daejeon, Korea
iskweon@kaist.ac.kr

## Abstract

Objects and their relationships are critical contents for image understanding. A scene graph provides a structured description that captures these properties of an image. However, reasoning about the relationships between objects is very challenging and only a few recent works have attempted to solve the problem of generating a scene graph from an image. In this paper, we present a method that improves scene graph generation by explicitly modeling inter-dependency among the entire object instances. We design a simple and effective *relational embedding module* that enables our model to jointly represent connections among all related objects, rather than focus on an object in isolation. Our method significantly benefits main part of the scene graph generation task: relationship classification. Using it on top of a basic Faster R-CNN, our model achieves state-of-the-art results on the Visual Genome benchmark. We further push the performance by introducing *global context encoding module* and *geometrical layout encoding module*. We validate our final model, LinkNet, through extensive ablation studies, demonstrating its efficacy in scene graph generation.

## 1  Introduction

Current state-of-the-art recognition models have made significant progress in detecting individual objects in isolation [9, 20]. However, we are still far from reaching the goal of capturing the interactions and relationships between these objects. While objects are the core elements of an image, it is often the relationships that determine the global interpretation of the scene. The deeper understating of visual scene can be realized by building a structured representation which captures objects and their relationships jointly. Being able to extract such graph representations have been shown to benefit various high-level vision tasks such as image search [13], question answering [2], and 3D scene synthesis [29].

In this paper, we address *scene graph generation*, where the objective is to build a visually-grounded scene graph of a given image. In a scene graph, objects are represented as nodes and relationships between them as directed edges. In practice, a node is characterized by an object bounding box with a category label, and an edge is characterized by a predicate label that connects two nodes as a *subject-predicate-object* triplet. As such, a scene graph is able to model not only what objects are in the scene, but how they relate to each other.

The key challenge in this task is to reason about inter-object relationships. We hypothesize that explicitly modeling inter-dependency among the entire object instances can improve a model's ability to infer their pairwise relationships. Therefore, we propose a simple and effective *relational embedding module* that enables our model to jointly represent connections among all related objects, rather than focus on an object in isolation. This significantly benefits main part of the scene graph generation task: relationship classification.

We further improve our network by introducing *global context encoding module* and *geometrical layout encoding module*. It is well known that fusing global and local information plays an important role in numerous visual tasks [8, 23, 39, 6, 40, 36]. Motivated by these works, we build a module that can provide contextual information. In particular, the module consists of global average pooling and binary sigmoid classifiers, and is trained for multi-label object classification. This encourages its intermediate features to represent all object categories present in an image, and supports our full model. Also, for the *geometrical layout encoding module*, we derive inspiration from the fact the most relationships in general are spatially regularized, implying that *subject-object* relative geometric layout can thus be a powerful cue for inferring the relationship in between. Our novel architecture results in our final model **LinkNet**, of which the overall architecture is illustrated in Fig. 1.

On the Visual Genome dataset, LinkNet obtains **state-of-the-art** results in scene graph generation tasks, revealing the efficacy of our approach. We visualize the weight matrices in relational embedding module and observe that inter-dependency between objects are indeed represented(see Fig. 2).

**Contribution.** Our main contribution is three-fold.

1. We propose a simple and effective *relational embedding module* in order to explicitly model inter-dependency among entire objects in an image. The *relational embedding module* improves the overall performance significantly.
2. In addition, we introduce *global context encoding module* and *geometrical layout encoding module* for more accurate scene graph generation.
3. The final network, LinkNet, has achieved new state-of-the art performance in scene graph generation tasks on the large-scale benchmark [16]. Extensive ablation studies demonstrate the effectiveness of the proposed network.

## 2 Related Work

**Relational Reasoning**    Relational reasoning has been explicitly modeled and adopted in neural networks. In the early days, most works attempted to apply neural networks to graphs, which are a natural structure for defining relations [11, 15, 24, 28, 1, 32]. Recently, the more efficient relational reasoning modules have been proposed [27, 30, 31]. Those can model dependency between the elements even with the non-graphical inputs, aggregating information from the feature embeddings at all pairs of positions in its input (e.g., pixels or words). The aggregation weights are automatically learned driven by the target task. While our work is connected to the previous works, an apparent distinction is that we consider object instances instead of pixels or words as our primitive elements. Since the objects have variations in scale/aspect ratio, we use ROI-align operation [9] to generate fixed 1D representations, easing the subsequent relation computations.

Moreover, relational reasoning of our model has a link to an attentional graph neural network. Similar to ours, *Chen et.al.* [4] uses a graph to encode spatial and semantic relations between regions and classes and passes information among them. To do so, they build a commonsense knowledge graph( *i.e.*adjacency matrix) from relationship annotations in the set. However, our approach does not require any external knowledge sources for the training. Instead, the proposed model generates soft-version of adjacency matrix(see Fig. 2) on-the-fly by capturing the inter-dependency among the entire object instances.

**Relationship Detection**    The task of recognizing objects and the relationships has been investigated by numerous studies in a various form. This includes detection of human-object interactions [7, 3], localization of proposals from natural language expressions [12], or the more general tasks of visual relationship detection [17, 25, 38, 5, 19, 37, 34, 41] and scene graph generation [33, 18, 35, 22].

Among them, scene graph generation problem has recently drawn much attention. The challenging and open-ended nature of the task lends itself to a variety of diverse methods. For example: fixing

the structure of the graph, then refining node and edge labels using iterative message passing [33]; utilizing associative embedding to simultaneously identify nodes and edges of graph and piece them together [22]; extending the idea of the message passing from [33] with additional RPN in order to propose regions for captioning and solve tasks jointly [18]; staging the inference process in three-step based on the finding that object labels are highly predictive of relation labels [35];

In this work, we utilize relational embedding for scene graph generation. It utilizes a basic self-attention mechanism [30] within the aggregation weights. Compared to previous models [33, 18] that have been proposed to focus on message passing *between* nodes and edges, our model explicitly reasons about the relations *within* nodes and edges and predicts graph elements in multiple steps [35], such that features of a previous stage provides rich context to the next stage.

# 3 Proposed Approach

## 3.1 Problem Definition

A *scene graph* is a topological representation of a scene, which encodes object instances, corresponding object categories, and relationships between the objects. The task of *scene graph generation* is to construct a scene graph that best associates its nodes and edges with the objects and the relationships in an image, respectively.

Formally, the graph contains a node set $\mathbf{V}$ and an edge set $\mathbf{E}$. Each node $v_i$ is represented by a bounding box $v_i^{bbox} \in \mathbb{R}^4$, and a corresponding object class $v_i^{cls} \in \mathbf{C_{obj}}$. Each edge $e_{i \to j} \in \mathbf{C_{rel}}$ defines a relationship *predicate* between the *subject* node $v_i$ and *object* node $v_j$. $\mathbf{C_{obj}}$ is a set of object classes, $\mathbf{C_{rel}}$ is a set of relationships. At the high level, the inference task is to classify objects, predict their bounding box coordinates, and classify pairwise relationship predicates between objects.

## 3.2 LinkNet

An overview of LinkNet is shown in Fig. 1. To generate a visually grounded scene graph, we need to start with an initial set of object bounding boxes, which can be obtained from ground-truth human annotation or algorithmically generated. Either cases are somewhat straightforward; In practice, we use a standard object detector, Faster R-CNN [26], as our bounding box model ($Pr(V^{bbox}|I)$). Given an image $I$, the detector predicts a set of region proposals $V^{bbox}$. For each proposal $v_i^{bbox}$, it also outputs a ROI-align feature vector $\mathbf{f_i^{RoI}}$ and an object label distribution $\mathbf{l_i}$.

We build upon these initial object features $\mathbf{f_i^{RoI}}$, $\mathbf{l_i}$ and design a novel scene graph generation network that consists of three modules. The first module is a *relational embedding module* that explicitly models inter-dependency among all the object instances. This significantly improves relationship classification($Pr(E_{i \to j}|I, V^{bbox}, V^{cls})$). Second, *global context encoding module* provides our model with contextual information. Finally, the performance of predicate classification is further boosted by our *geometric layout encoding*.

In the following subsections, we will explain how each proposed modules are used in two main steps of scene graph generation: object classification, and relationship classification.

## 3.3 Object Classification

### 3.3.1 Object-Relational Embedding

For each region proposal, we construct a relation-based representation by utilizing the object features from the underlying RPN: the ROI-aligned feature $\mathbf{f_i^{RoI}} \in \mathbb{R}^{4096}$ and embedded object label distribution $\mathbf{K_0 l_i} \in \mathbb{R}^{200}$. $\mathbf{K_0}$ denotes a parameter matrix that maps the distribution of predicted classes, $\mathbf{l_i}$, to $\mathbb{R}^{200}$. In practice, we use an additional image-level context features $\mathbf{c} \in \mathbb{R}^{512}$, so that each object proposal is finally represented as a concatenated vector $\mathbf{o_i} = (\mathbf{f_i^{RoI}}, \mathbf{K_0 l_i}, \mathbf{c})$. We detail on the global context encoding in Sec. 3.3.2.

Then, for a given image, we can obtain N object proposal features $\mathbf{o_{i=1,...,N}}$. Here, we consider *object-relational embedding* $\mathbf{R}$ that computes the response for one object region $\mathbf{o_i}$ by attending to the features from all N object regions. This is inspired by the recent works for relational reasoning [27, 30, 31]. Despite the connection, what makes our work distinctive is that we consider object-level

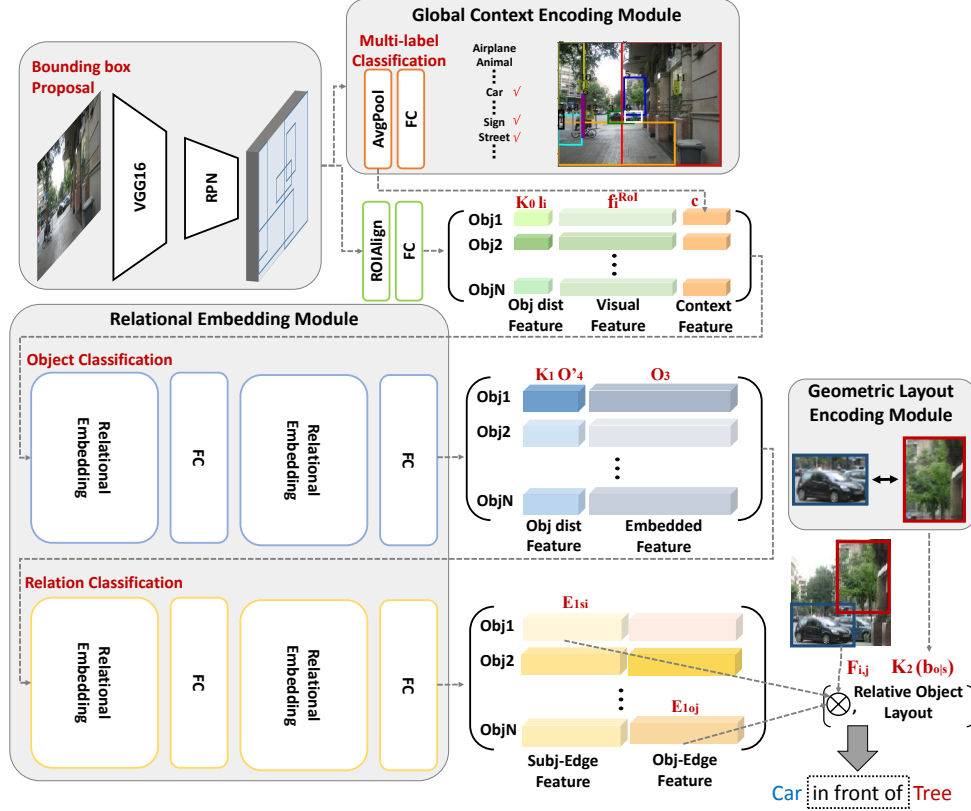

Figure 1: **The overview of LinkNet.** The model predicts graph in three steps: bounding box proposal, object classification, and relationship classification. The model consists of three modules: global context encoding module, relational embedding module, and geometric layout encoding module. Best viewed in color.

instances as our primitive elements, whereas the previous methods operate on pixels [27, 31] or words [30].

In practice, we stack all the object proposal features to build a matrix $\mathbf{O_0} \in \mathbb{R}^{\mathbf{N} \times 4808}$, from where we can compute a relational embedding matrix $\mathbf{R_1} \in \mathbb{R}^{\mathbf{N} \times \mathbf{N}}$. Then, the relation-aware embedded features $\mathbf{O_2} \in \mathbb{R}^{\mathbf{N} \times \mathbf{256}}$ are computed as:

$$\mathbf{R_1} = \text{softmax}((\mathbf{O_0 W_1})(\mathbf{O_0 U_1})^{\mathbf{T}}) \in \mathbb{R}^{\mathbf{N} \times \mathbf{N}}, \tag{1}$$

$$\mathbf{O_1} = \mathbf{O_0} \oplus fc_0((\mathbf{R_1}(\mathbf{O_0 H_1}))) \in \mathbb{R}^{\mathbf{N} \times \mathbf{4808}}, \tag{2}$$

$$\mathbf{O_2} = fc_1(\mathbf{O_1}) \in \mathbb{R}^{\mathbf{N} \times \mathbf{256}}, \tag{3}$$

where $\mathbf{W_1}$, $\mathbf{U_1}$ and $\mathbf{H_1}$ are parameter matrices that map the object features, $\mathbf{O_0}$ to $\mathbb{R}^{\mathbf{N} \times \frac{4808}{r}}$, here we found setting hyper-parameter r as 2 produces best result from our experiment. The softmax operation is conducted in row-wise, constructing an embedding matrix. $fc_0$ and $fc_1$ are a parameter matrices that map its input feature of $\mathbb{R}^{N \times \frac{4808}{r}}$ to $\mathbb{R}^{N \times 4808}$, and $\mathbb{R}^{N \times 4808}$ to an embedding space $\mathbb{R}^{N \times 256}$, respectively. $\oplus$ denotes a element-wise summation, allowing an efficient training overall due to residual learning mechanism [10]. The resulting feature $\mathbf{O_2}$ again goes through a similar relational embedding process, and is eventually embedded into object label distribution $\mathbf{O_4} \in \mathbb{R}^{\mathbf{N} \times \mathbf{C_{obj}}}$ as:

$$\mathbf{R_2} = \text{softmax}((\mathbf{O_2 W_2})(\mathbf{O_2 U_2})^{\mathbf{T}}) \in \mathbb{R}^{\mathbf{N} \times \mathbf{N}}, \tag{4}$$

$$\mathbf{O_3} = \mathbf{O_2} \oplus fc_2((\mathbf{R_2}(\mathbf{O_2 H_2}))) \in \mathbb{R}^{\mathbf{N} \times \mathbf{256}}, \tag{5}$$

$$\mathbf{O_4} = fc_3(\mathbf{O_3}) \in \mathbb{R}^{\mathbf{N} \times \mathbf{C_{obj}}}, \tag{6}$$

where $\mathbf{W_2}$, $\mathbf{U_2}$ and $\mathbf{H_2}$ map the object features, $\mathbf{O_2}$ to $\mathbb{R}^{\mathbf{N} \times \frac{256}{r}}$. The softmax operation is conducted in row-wise, same as above. $fc_2$ and $fc_3$ are another parameter matrices that map the intermediate features into $\mathbb{R}^{\mathbf{N} \times \frac{256}{r}}$ to $\mathbb{R}^{\mathbf{N} \times 256}$, and $\mathbb{R}^{\mathbf{N} \times 256}$ to $\mathbb{R}^{\mathbf{N} \times C_{obj}}$, respectively. Finally, the $\mathbf{C_{obj}}$-way object classification $Pr(V^{cls}|I, V^{bbox})$ is optimized on the resulting feature $\mathbf{O_4}$ as:

$$\hat{V}^{cls} = \mathbf{O_4}, \tag{7}$$

$$\mathcal{L}_{obj\_cls} = -\sum \quad V^{cls} \log(\hat{V^{cls}}). \tag{8}$$

### 3.3.2 Global Context Encoding

Here we describe the *global context encoding module* in detail. This module is designed with the intuition that knowing contextual information in prior may help inferring individual objects in the scene.

In practice, we introduce an auxiliary task of multi-label classification, so that the intermediate features $\mathbf{c}$ can encode all kinds of objects present in an image. More specifically, the *global context encoding* $\mathbf{c} \in \mathbb{R}^{\mathbf{512}}$ is taken from an average pooling on the RPN image features($\mathbb{R}^{512 \times H \times W}$), as shown in Fig. 1. This feature $\mathbf{c}$ is concatenated with the initial image features ($\mathbf{f_i}$, $\mathbf{K_0 l_i}$) as explained in Sec. 3.3.1, and supports scene graph generation performance as we will demonstrate in Sec. 4.2. After one parameter matrix , $\mathbf{c}$ becomes multi-label distribution $\hat{\mathbf{M}} \in (\mathbf{0}, \mathbf{1})^{\mathbf{C_{obj}}}$, and multi-label object classification (gce loss) is optimized on the ground-truth labels $\mathbf{M} \in [\mathbf{0}, \mathbf{1}]^{\mathbf{C_{obj}}}$ as:

$$\mathcal{L}_{gce} = -\sum \sum_{c=1}^{\mathbf{C_{obj}}} \mathbf{M_c} \log(\hat{\mathbf{M}}_{\mathbf{c}}). \tag{9}$$

## 3.4 Relationship Classification

### 3.4.1 Edge-Relational Embedding

After the object classification, we further construct relation-based representations suitable for relationship classification. For this, we apply another sequence of *relational embedding modules*. In particular, the output of the previous *object-relational embedding module* $\mathbf{O_4} \in \mathbb{R}^{\mathbf{N} \times \mathbf{C_{obj}}}$, and the intermediate feature $\mathbf{O_3} \in \mathbb{R}^{\mathbf{N} \times \mathbf{256}}$ are taken as inputs as:

$$\mathbf{O_4'} = \mathbf{argmax}(\mathbf{O_4}) \in \mathbb{R}^{\mathbf{N} \times \mathbf{C_{obj}}}, \tag{10}$$

$$\mathbf{E_0} = (\mathbf{K_1 O_4'}, \mathbf{O_3}) \in \mathbb{R}^{\mathbf{N} \times (\mathbf{200} + \mathbf{256})}, \tag{11}$$

where the argmax is conducted row-wise and produces an one-hot encoded vector $\mathbf{O_4'}$ which is then mapped into $\mathbb{R}^{N \times 200}$ by a parameter matrix $\mathbf{K_1}$. Then, similar embedding operations as in Sec. 3.3.1 are applied on $\mathbf{E_0}$, resulting in embedded features $\mathbf{E_1} \in \mathbb{R}^{\mathbf{N} \times \mathbf{8192}}$, where the half of the channels(4096) refers to *subject* edge features and its counterpart refers to *object* (see Fig. 1).

For each possible $\mathbf{N^2 - N}$ edges, say between $\mathbf{v_i}$ and $\mathbf{v_j}$, we compute the probability the edge will have label $e_{i \rightarrow j}$ (including the background). We operate on $\mathbf{E_1}$ and an embedded features from the union region of $i$-th and $j$-th object regions, $\mathbf{F} = \{ \mathbf{f_{i,j}} \mid i \in (1, 2, ...N), j \in (1, 2, ..., \mathbf{N}), \mathbf{j} \neq \mathbf{i} \} \in \mathbb{R}^{\mathbf{N(N-1)} \times 4096}$ as:

$$\mathbf{G_{0_{ij}}} = (\mathbf{E_{1_{si}}} \otimes \mathbf{E_{1_{oj}}} \otimes \mathbf{F_{ij}}) \in \mathbb{R}^{\mathbf{4096}}, \tag{12}$$

$$\mathbf{G_1} = (\mathbf{G_0}, \mathbf{K_2}(\mathbf{b_{o|s}})) \in \mathbb{R}^{\mathbf{N(N-1)} \times (\mathbf{4096} + \mathbf{128})}, \tag{13}$$

$$\mathbf{G_2} = fc_4(\mathbf{G_1}) \in \mathbb{R}^{\mathbf{N(N-1)} \times \mathbf{C_{rel}}}. \tag{14}$$

We combine *subject* edge features, *object* edge features and union image representations by low-rank outer product [14]. $\mathbf{b_{o|s}}$ denotes relative geometric layout which is detailed in Sec. 3.4.2. It is embedded into $\mathbb{R}^{\mathbf{N(N-1)} \times \mathbf{128}}$ by a parameter matrix $\mathbf{K_2}$. A parameter matrix $fc_4$ maps the intermediate features $\mathbf{G_1} \in \mathbb{R}^{\mathbf{N(N-1)} \times 4224}$ into $\mathbf{G_2} \in \mathbb{R}^{\mathbf{N(N-1)} \times \mathbf{C_{rel}}}$

| Methods | Predicate Classification | | | Scene Graph Classification | | | Scene Graph Detection | | |
|---|---|---|---|---|---|---|---|---|---|
| | R@20 | R@50 | R@100 | R@20 | R@50 | R@100 | R@20 | R@50 | R@100 |
| VRD [21] | | 27.9 | 35.0 | | 11.8 | 14.1 | | 0.3 | 0.5 |
| MESSAGE PASSING [33] | | 44.8 | 53.0 | | 21.7 | 24.4 | | 3.4 | 4.2 |
| ASSOC EMBED [22] | 47.9 | 54.1 | 55.4 | 18.2 | 21.8 | 22.6 | 6.5 | 8.1 | 8.2 |
| MOTIFNET [35] | 58.5 | 65.2 | 67.1 | 32.9 | 35.8 | 36.5 | 21.4 | 27.2 | **30.3** |
| **LinkNet** | **61.8** | **67.0** | **68.5** | **38.3** | **41** | **41.7** | **22.3** | **27.4** | 30.1 |

Table 1: The table shows our model achieves state-of-the-art result in Visual Genome benchmark [16]. Note that the **Predicate Classification** and **Scene Graph Classification** tasks assume exactly same perfect detector across the methods, and evaluate how well the each models predict object labels and their relations, while **Scene Graph Detection** task takes a customized pre-trained detector and performs subsequent tasks.

Finally, the $\mathbf{C_{rel}}$-way relationship classification $Pr(E_{i \to j}|I, V^{bbox}, V^{cls})$ is optimized on the resulting feature $\mathbf{G_2}$ as:

$$\hat{E_{i \to j}} = \mathbf{G_2}, \tag{15}$$

$$\mathcal{L}_{rel\_cls} = -\sum_{i=1}^{\mathbf{N}} \sum_{j \neq i} E_{i \to j} \log(\hat{E}_{i \to j}). \tag{16}$$

### 3.4.2 Geometric Layout Encoding

We hypothesize that relative geometry between the *subject* and *object* is a powerful cue for inferring the relationship between them. Indeed, many *predicates* have straightforward correlation with the *subject-object* relative geometry, whether they are geometric (e.g., *'behind'*), possessive (e.g.,*'has'*), or semantic (e.g.,*'riding'*).

To exploit this cue, we encode the relative location and scale information as :

$$\mathbf{b_{o|s}} = \left(\frac{\mathbf{x_o} - \mathbf{x_s}}{\mathbf{w_s}}, \frac{\mathbf{y_o} - \mathbf{y_s}}{\mathbf{h_s}}, \log(\frac{\mathbf{w_o}}{\mathbf{w_s}}), \log(\frac{\mathbf{h_o}}{\mathbf{h_s}})\right), \tag{17}$$

where $\mathbf{x}, \mathbf{y}, \mathbf{w}$, and $\mathbf{h}$ denote the x,y-coordinates, width, and height of the object proposal, and the subscripts $\mathbf{o}$ and $\mathbf{s}$ denote *object* and *subject*, respectively. we embed $\mathbf{b_{o|s}}$ to a feature in $\mathbb{R}^{\mathbf{N} \times 128}$ and concatenate this with the *subject-object* features as in Eq. (13).

### 3.5 Loss

The whole network can be trained in an end-to-end manner, allowing the network to predict object bounding boxes, object categories, and relationship categories sequentially (see Fig. 1). Our loss function for an image is defined as:

$$\mathcal{L}_{final} = \mathcal{L}_{obj\_cls} + \lambda_1 \mathcal{L}_{rel\_cls} + \lambda_2 \mathcal{L}_{gce}. \tag{18}$$

By default, we set $\lambda_1$ and $\lambda_2$ as 1, and thus all the terms are equally weighted.

## 4 Experiments

We conduct experiments on Visual Genome benchmark [16].

### 4.1 Quantitative Evaluation

Since the current work in scene graph generation is largely inconsistent in terms of data splitting and evaluation, we compared against papers [21, 33, 22, 35] that followed the original work [33]. The experimental results are summarized in Table. 1.

The LinkNet achieves new **state-of-the-art** results in Visual Genome benchmark [16], demonstrating its efficacy in identifying and associating objects. For the scene graph classification and predicate classification tasks, our model outperforms the strong baseline [35] by a large margin. Note that predicate classification and scene graph classification tasks assume the same perfect detector across the methods, whereas scene graph detection task depends on a customized pre-trained detector.

| Independent Variables | Value | Scene Graph Classification | | |
|---|---|---|---|---|
| | | R@20 | R@50 | R@100 |
| Number of REM | 1 | 37.7 | 40.4 | 41 |
| | **ours(2)** | **38.3** | **41** | **41.7** |
| | 3 | 37.9 | 40.6 | 41.3 |
| | 4 | 38 | 40.7 | 41.4 |
| Reduction ratio (r) | 1 | 38 | 40.9 | 41.6 |
| | **ours(2)** | **38.3** | **41** | **41.7** |
| | 4 | 38.2 | 41 | 41.6 |
| | 8 | 37.7 | 40.5 | 41.2 |

a Experiments on hyperparams.

| Exp | Operation | | Scene Graph Classification | | |
|---|---|---|---|---|---|
| | argmax(O4) | concat(O3) | R@20 | R@50 | R@100 |
| 1 | ✓ | | 37.3 | 39.8 | 40.6 |
| 2 | | ✓ | 38 | 40.7 | 41.4 |
| Ours | ✓ | ✓ | **38.3** | **41** | **41.7** |

b Design-choices in constructing E0.

Table 2: **(a)** includes experiments for the optimal value for the two hyper parameters; **(b)** includes experiments to verify the effective design choices of constructing E0

| Exp | Proposed | | | Row-wise | | Similarity | | Scene Graph Classification | | |
|---|---|---|---|---|---|---|---|---|---|---|
| | REM | GLEM | GCEM | Softmax | Sigmoid | Dot prod | Eucli | R@20 | R@50 | R@100 |
| 1 | ✓ | | | ✓ | | ✓ | | 37.4 | 40.0 | 40.8 |
| 2 | ✓ | ✓ | | ✓ | | ✓ | | 37.9 | 40.4 | 41.2 |
| 3 | ✓ | | ✓ | ✓ | | ✓ | | 38.0 | 40.6 | 41.3 |
| 4 | ✓ | ✓ | ✓ | | ✓ | ✓ | | 37.7 | 40.3 | 41 |
| 5 | ✓ | | | ✓ | | | ✓ | 37.2 | 40.0 | 40.7 |
| 6 | ✓ | ✓ | ✓ | ✓ | | | ✓ | 37.9 | 40.7 | 41.4 |
| Ours | ✓ | ✓ | ✓ | ✓ | | ✓ | | **38.3** | **41** | **41.7** |

| | R@100 | |
|---|---|---|
| predicate | w. GLEM | w.o GLEM |
| using | 0.269 | 0.000 |
| carrying | 0.246 | 0.118 |
| riding | 0.249 | 0.138 |
| behind | 0.341 | 0.287 |
| at | 0.072 | 0.040 |
| in front of | 0.094 | 0.069 |
| has | 0.495 | 0.473 |
| wearing | 0.488 | 0.468 |
| on | 0.570 | 0.551 |
| sitting on | 0.088 | 0.070 |

Table 3: The **left table** shows ablation studies on the final model. The **right table** summarizes top-10 predicates with highest recall increase in scene graph classification with the use of geometric layout encoding module. **REM**, **GLEM**, **GCEM** denotes Relational Embedding Module, Geometric Layout Encoding Module, and Global Context Encoding Moudle respectively.

## 4.2 Ablation Study

In order to evaluate the effectiveness of our model, we conduct four ablation studies based on the scene graph classification task as follows. Results of the ablation studies are summarized in Table. 2 and Table. 3.

**Experiments on hyperparameters** The first row of Table. 2a shows the results of *more relational embedding modules*. We argue that multiple modules can perform multi-hop communication. Messages between all the objects can be effectively propagated, which is hard to do via standard models. However, too many modules can arise optimization difficulty. Our model with two REMs achieved the best results. In second row of Table. 2a, we compare performance with four different *reduction ratios*. The reduction ratio determines the number of channels in the module, which enables us to control the capacity and overhead of the module. The reduction ratio 2 achieves the best accuracy, even though the reduction ratio 1 allows higher model capacity. We see this as an over-fitting since the training losses converged in both cases. Overall, the performance drops off smoothly across the reduction ratio, demonstrating that our approach is robust to it.

**Design-choices in constructing** $E_0$ Here we construct an input($E_0$) of edge-relational embedding module by combining an object class representation($O_4'$) and a global contextual representation($O_3$). The operations are inspired by the recent work [35] that contextual information is critical for the relationship classification of an objects. To do so, we turn $O_4$ of object label probabilities into one-hot vectors via an argmax operation(committed to a specific object class label) and we concatenate it with an output($O_3$) which passed through the relational embedding module(contextualized representation). As shown in the Table. 2b, we empirically confirm that both operations contribute to the performance boost.

**The effectiveness of proposed modules.** We perform an ablation study to validate our modules in the network, which are relation embedding module, geometric layout encoding module, and global context encoding module. We remove each module to verify the effectiveness of utilizing all the proposed modules. As shown in Exp 1, 2, 3, and Ours, we can clearly see the performance improvement when we use all the modules jointly. This shows that each module plays a critical role together in inferring object labels and their relationships. Note that **Exp 1** already achieves

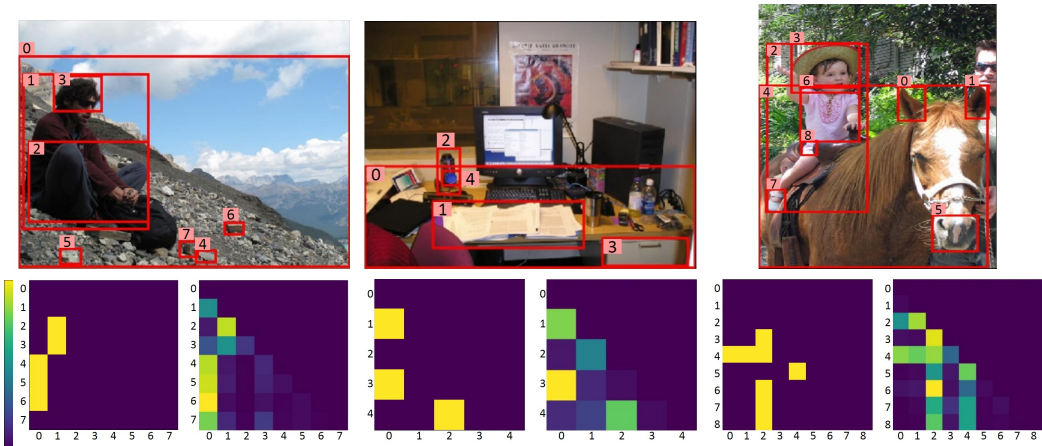

Figure 2: **Visualization of relational embedding matrices.** For each example, the first row shows ground-truth object regions. The left and right column of the second row show ground-truth relations (binary, 1 if present, 0 otherwise), and the weights of our relational embedding matrix, respectively. Note how the relational embedding relates the objects with a real connection, compared to those in none-relationship.

state-of-the-art result, showing that utilizing *relational embedding module* is crucial while the other modules further boost performance.

**The effectiveness of GLEM.**  We conduct additional analysis to see how the network performs with the use of *geometric layout encoding module*. We select top-10 predicates with highest recall increase in scene graph classification task. As shown in right side of Table. 3, we empirically confirm that the recall value of geometrically related predicates are significantly increased, such as *using*, *carrying*, *riding*. In other words, predicting predicates which have clear *subject-object* relative geometry, was helped by the module.

**Row-wise operation methods.**  In this experiment, we conduct an ablation study to compare row-wise operation methods in relational embedding matrix: softmax and sigmoid; As we can see in Exp 4 and Ours, softmax operation which imposes competition along the row dimension performs better, implying that explicit attention mechanism [30] which emphasizes or suppresses relations between objects helps to build more informative embedding matrix.

**Relation computation methods.**  In this experiment, we investigate two commonly used relation computation methods: dot product and euclidean distance. As shown in Exp 1 and 5, we observe that dot-product produces slightly better result, indicating that relational embedding behavior is crucial for the improvement while it is less sensitive to computation methods. Meanwhile, Exp 5 and 6 shows that even we use euclidean distance method, *geometric layout encoding module* and *global context encoding module* further improves the overall performance, again showing the efficacy of the introduced modules.

### 4.3   Qualitative Evaluation

**Visualization of relational embedding matrix**   We visualize our relational embedding of our network in Fig. 2. For each example, the bottom-left is the ground-truth binary triangular matrix where its entry is filled as: $(i, j \,|i < j) = \mathbf{1}$ only if there is a non-background relationship(in any direction) between the $i$-th and $j$-th instances, and $\mathbf{0}$ otherwise. The bottom-right is the trained weights of an intermediate relational embedding matrix (Eq. (4)), folded into a triangular form. The results show that our relational embedding represents inter-dependency among all object instances, being consistent with the ground-truth relationships. To illustrate, in the first example, the ground-truth matrix refers to the relationships between the *'man'*(1) and his body parts(2,3); and the *'mountain'*(0) and the *'rocks'*(4,5,6,7), which are also reasonably captured in our relational embedding matrix. Note that our model infers relationship correctly even there exists missing ground-truths such as cell(7,0) due to sparsity of annotations in Visual Genome dataset. Indeed, our *relational embedding module*

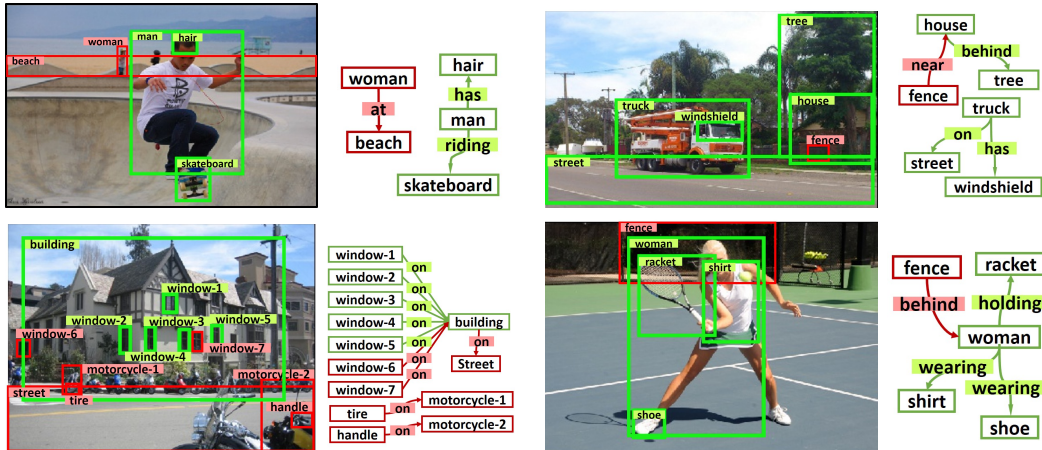

Figure 3: **Qualitative examples of our scene graph detection results.** Green boxes and edges are correct predictions, and red boxes and edges are false negatives.

plays a key role in scene graph generation, leading our model to outperform previous state-of-the-art methods.

**Scene graph detection**     Qualitative examples of scene graph detection of our model are shown in Fig. 3. We observe that our model properly induces scene graph from a raw image.

# 5    Conclusion

We addressed the problem of generating a scene graph from an image. Our model captures global interactions of objects effectively by proposed *relational embedding module*. Using it on top of basic Faster R-CNN system significantly improves the quality of node and edge predictions, achieving state-of-the-art result. We further push the performance by introducing *global context encoding module* and *geometric layout encoding module*, constructing a LinkNet. Through extensive ablation experiments, we demonstrate the efficacy of our approach. Moreover, we visualize relational embedding matrix and show that relations are properly captured and utilized. We hope LinkNet become a generic framework for scene graph generation problem.

**Acknowledgements**     This research is supported by the Study on Deep Visual Understanding funded by the Samsung Electronics Co., Ltd (Samsung Research)

## Footnotes

\*Both authors have equally contributed

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
