[Reviews · NeurIPS 2018]

Reviewer 1



This paper tackles the problem of scene graph estimation, introducing a relational embedding module to process object features and explicitly compare all pairs of objects to improve relationship prediction. Furthermore, the authors show that introduction of features that capture global context and geometric relations further boosts performance. Strengths: - Each contributed component is well motivated and brings something independent to the table. I think the design of the relational embedding module is pretty reasonable. It directly compares all pairs of object features and then residually updates each of their representations. Further including global image features and explicit spatial features results in a well-rounded feature pipeline for scene graph estimation. - The ablations in Table 2 are appropriate, and the results on Visual Genome are strong Weaknesses: - L107 claims that the relational embedding module significantly improves object classification, but we never see any results to back up this claim. It would strengthen the case of the paper to report the improvement on object classification directly in addition to the overall scene graph results. - There are a number of design decisions that feel a bit arbitrary. Obviously it is not possible to exhaustively go through and perform an ablation for every single little detail, but some choices seem like they could have used further discussion. For example, in 3.3.1 why is the relational embedding module used twice on the object features? How important is the choice of the hyperparameter r? Why do an argmax over O_4 and convert it to a one-hot representation, what about using it as is and preserving the distribution as is done with l_i (L118)? - There is a lot going on in Figure 1, and to tell the truth, before reading the paper I couldn’t really make much sense of it. Even after reading the paper it is a bit difficult to parse. Rather than fitting everything into one big figure, it might make more sense to break visualizations down into separate stages. Overall assessment: While I would not say the contributions are necessarily big steps away from existing methods, the overall pipeline is well thought out and laid out clearly enough for the reader. Experiments justify the inclusion of the various proposed components, and final performance is better than the current state of the art. As such, I am leaning towards acceptance. Additional questions/comments: - In the relational embedding eqs, why distinguish between multiplying by a parameter matrix and passing through a fully-connected layer? Does a bias or lack thereof make a difference? - A figure illustrating the sequence of operations that make up the relational embedding module (eqs 1,2,3 and then again in 4,5,6) could be helpful since this is a key operation repeated several times. - Some of the tables/figures are far from the corresponding text (the ablation results for example) - How many object proposals are usually considered by the relational embedding module - what is a typical N? Update: The authors provide a number of good ablations and clarifications in their rebuttal, strengthening my recommendation for acceptance. I will mention that I do not totally agree with L34 of the rebuttal. It is difficult to disentangle what contributes to a final number, and directly reporting object classification performance would still be informative and necessary to back up the claim that object classification is improved with the REM.

Reviewer 2



Summary ------------- This paper proposes an attentional graph neural network approach for scene graph classification (classifying the objects in a scene, as well as the relationships between those objects) and applies it to the Visual Genome dataset, achieving state-of-the art performance with significant improvements particularly on the scene graph classification tasks. Overall, I think this approach seems very promising and the results are quite impressive (though I should say I am less familiar with the literature on scene graph classification specifically). The architecture used in the paper is not particularly novel as graph neural networks have (1) previously been tried and (2) the paper mostly just borrows its architecture from the Transformer architecture (see below), but I think the impressive results can make up for a lack of novelty. However, I did find the paper somewhat difficult to read and it took me several passes through Section 3 to understand exactly what was going on. Additionally, I found the experiments to be a bit weak and some of the architectural choices to be somewhat arbitrary and not justified. Quality --------- Some of the choices of the architecture seem somewhat arbitrary and it's not clear to me their significance. For example, why is it necessary to turn O_4 into one-hot vectors via an argmax? Or, why is O_3 used in constructing E_0? Does these choices really change performance? I appreciate the ablation experiments which show that the Global Context Embedding, Geometric Layout Encoding, and Relational Embedding are all important aspects, but I feel these experiments could have gone further. For example, the relational embedding is used four times: twice for node classification and twice for edge classification. Is it needed for both node and edge classification, or is it sufficient to just use it e.g. for node classification? And how much of an impact is there in using two stacked relational embeddings rather than one (or three)? Moreover, there are no ablation experiments which completely remove the relational embedding (but e.g. leave in the global context or the geometric layout encoding), making it difficult to judge how much that is improving things over a baseline. Clarity -------- I found Section 3 hard to follow, with inconsistent and sometimes confusing notation. The following points were the hardest for me to follow: - Sometimes the text specifies a variable name for a parameter matrix (e.g. W_1, U_1), and sometimes it uses an fc layer (e.g. fc_0, fc_1). I take it the difference is that the parameter matrices don't include bias terms, but I think it would be far easier to understand if the paper just used fc functions everywhere and specified somewhere else (e.g. in a footnote, or in the appendix) which fc terms used a bias and which did not. - I would strongly recommend a revision include (for example) an algorithm box which makes it clearer exactly how the computation is structured (this could even be in the appendix). - I found Figure 1 to be confusing, especially since there are terms used in the figure which aren't defined anywhere but the text (ideally, the figure should be interpretable in isolation). For example, O_3 and O_4 are not defined in the figure. I think the figure could be redone to be more explicit about where the different terms come from (I think there is a disproportionate amount of whitespace in the relational embedding and FC boxes, so these could be shrunk to make room for more detail). - Another detail which wasn't clear to me is where the features f_{i,j} (which are used in Eq. 12) come from. Are these the same ROI features introduced in 3.3.1? If so, how are they indexed by j? I thought f_{i} was of dimensionality 4096. Originality ------------- There are two forms of novelty in this paper: first, applying the idea of attentional graph neural networks to scene graph classification, and second, including auxiliary losses and computations in the global context embedding and the geometric layout encoding. However, the architecture of the relational embedding that is used (Eqs. 1-3 and 4-5) is almost exactly that from the Transformer architecture (Vaswani et al., 2017). Moreover, the idea of using (non-attentional) graph neural networks in the context of scene graph classification was previously tried by Xu et al. (2017). Thus, while the results are impressive, there does not seem to be much innovation in the architecture. Some other recent work has also used graph neural networks for just region classification and may be worth discussing and comparing to, see: Chen, X., Li, L., Fei-Fei, L., and Gupta, A. (2018). Iterative visual reasoning beyond convolutions. CVPR 2018. Significance ---------------- As far as graph neural networks go, it is very interesting to see them applied more widely to more naturalistic data like raw images and I think this paper is significant in that regard (even if it is not the absolute first to do so). More significant are the results that the method achieves on the Visual Genome dataset, which seem to significantly push the state of the art. Edit after author response: I have read the other reviews and the author response and am quite satisfied with the addition of several new ablation experiments. I do wish the authors had attempted to explain why they made the choices that they did about E_0 and O_4 in the first place (and hope that they will in the final revision), but it is good to see the empirical experiments showing those choices do matter. I also wish that the authors had explained a bit further how they see their work as different from previous approaches in the rebuttal, as suggested by R3, and hope that they will do so in the final version. All that said, I think the results are strong enough and the experiments now thorough enough that I have increased my score to a 7.

Reviewer 3



Summary: This paper presents a framework (referred as to LinkNet) for scene graph generation, i.e. the task of generating a visually grounded scene graph of an image. In a scene graph objects are represented as nodes and relationships between them as directed edges. State-of-the-art scene graph generation results are reported on VisualGenome. Strengths: - The proposed approach seems really effective. It combines three main components: i) a global-context encoding module, ii) a relational embedding module, iii) a geometric layout encoding module. I believe the overall approach is going in an very interesting direction by trying to encode all the main ingredients that can lead to scene graph generation. This is achieved in an end-to-end architecture. - Experimental results are very interesting. The proposed model outperforms recent related works on predicate classification, scene graph classification, and scene graph detection, on the large VisualGenome dataset. Weaknesses: - Presentation could be improved. In general the method is well presented and I am somewhat confident that the model could be implemented by the readers. However, although very effective, the approach seems incremental and the authors should better clarify their main contributions. For example, the authors claim that comparing to previous methods (e.g. [17,32]) which focus on message passing between nodes and edges, the proposed model explicitly reasons about the relations within nodes and edges and predicts graph elements in multiple steps [34]. I think the author should clarify what are the main differences w.r.t. to both [17,32] and [34]. - The relational embedding module explicitly models inter-dependency among all the object instances. This seems a major difference with respect to similar previous works (see above). The authors claim that this significantly improve both object classification and relationship classification. I ma not fully convinced that this has been clearly highlighted by the experimental results. Questions and other minor comments: - What those it mean that the Predicate Classification and Scene Graph Classification tasks assume exactly same perfect detector across the methods (Table 1)? Are you using ground-truth data instead of the object detection outputs? - Ablation studies and qualitative evaluations are very interesting. However, I think Figure 2 is quite hard to follow. Maybe you can just report three examples (instead of six) and try to visualise the relationships encoded by the relational embedding matrix. Section 4.3 is probably ok, but I think it takes a while to parse those examples. - Another comment regarding Fig.2 and Section 4.3: it seems that the ground truth matrix for the first example is wrong. The cell (7,0) should be also in yellow since there are four instances of 'rocks' (4,5,6,7). Moreover, I believe you should also report the color bars in order to better understand the effectiveness of your relational embedding matrices.